# 17R/S-Benzo-RvD1, a synthetic resolvin D1 analogue, attenuates neointimal hyperplasia in a rat model of acute vascular injury

**Alexander S. Kim**[ID][☯], **Evan C. Werlin**[☯†], **Hideo Kagaya, Mian Chen, Bian Wu, Giorgio Mottola, Masood Jan, Michael S. Conte**[ID]*

Department of Surgery and Cardiovascular Research Institute, UCSF, San Francisco, California, United States of America

☯ These authors contributed equally to this work.
† Deceased.
* michael.conte2@ucsf.edu

**Data Availability Statement:** All relevant data are within the manuscript and its supporting information files.

## Abstract

### Background

Persistent inflammation following vascular injury drives neointimal hyperplasia (NIH). Specialized lipid mediators (SPM) mediate resolution which attenuates inflammation and downstream NIH. We investigated the effects of a synthetic analogue of resolvin D1 (RvD1) on vascular cells and in a model of rat carotid angioplasty.

### Methods

Human venous VSMC and endothelial cells (EC) were employed in migration, cell shape, toxicity, proliferation and p65 nuclear translocation assays. Murine RAW 264.7 cells were utilized to test the effect of pro-resolving compounds on phagocytic activity. A model of rat carotid angioplasty was used to evaluate the effects of 17R/S-benzo-RvD1 (benzo-RvD1) and 17R-RvD1 applied to the adventitia via 25% Pluronic gel. Immunostaining was utilized to examine Ki67 expression and leukocyte recruitment. Morphometric analysis was performed on arteries harvested 14 days after injury.

### Results

Exposure to benzo-RvD1 attenuated PDGF- stimulated VSMC migration across a range of concentrations (0.1–100 nM), similar to that observed with 17R-RvD1. Pre-treatment with either Benzo-RvD1 or 17R-RvD1 (10, 100nM) attenuated PDGF-BB-induced VSMC cytoskeletal changes to nearly baseline dimensions. Benzo-RvD1 demonstrated modest antiproliferative activity on VSMC and EC at various concentrations, without significant cytotoxicity. Benzo-RvD1 (10nM) inhibited p65 nuclear translocation in cytokine-stimulated EC by 21% (p<0.05), similar to 17R-RvD1. Consistent with pro-resolving activities of other SPM, both 17R-RvD1 and benzo-RvD1 increased the phagocytic activity of RAW 264.7 cells against S. Aureus and Zymosan particles. There were no significant differences in Ki-67 or CD45 staining observed on day 3 after angioplasty. Periadventitial treatment with benzo-

**Funding:** MSC NIH VITA HHSN268201400005C U. S. National Institutes of Health Vascular Interventions/Innovations and Therapeutic Advances Program https://www.nhlbi.nih.gov/science/vascular-interventionsinnovations-and-therapeutic-advances-vita-program MSC NHLBI R01 HL119508 U.S. National Institutes of Health, National Heart, Lung, and Blood Institute Grants https://www.nhlbi.nih.gov/ ASK NIH R25 23856 U. S. National Institutes of Health Research Training and Career Development https://researchtraining.nih.gov/programs/research-education/r25 The funders had no role in study design, data collection and analysis, decision to publish, or preparation of the manuscript.

**Competing interests:** MSC - co-Founder of VasaRx and co-inventor of IP related to this work with the Regents of University of California and Brigham and Women's Hospital. This does not alter our adherence to PLOS-One policies on sharing data and materials.

RvD1 reduced carotid neointimal area at 14 days compared to control (0.08 mm$^2$ v. 0.18 mm$^2$; p<0.05), with similar efficacy to 17R-RvD1.

## Conclusions

17R/S-benzo-RvD1 and 17R-RvD1 exhibit similar pro-resolving and anti-migratory activity in cell-based assays, and both compounds attenuated NIH following acute arterial injury in rats. Further studies of the mechanisms of resolution following vascular injury, and the translational potential of SPM analogues, are indicated.

## Introduction

Peripheral arterial disease (PAD) due to atherosclerosis leads to impaired circulation to various areas of the body, most commonly the lower extremities [1]. PAD causes significant patient morbidity, leading to pain, chronic wounds, and limb amputation [2]. Approximately 8–10 million people in the US [3] and greater than 200 million worldwide [4, 5] suffer from PAD, and this translates to billions of dollars of healthcare utilization [6]. While vascular interventions for PAD (e.g. angioplasty, stenting, atherectomy, bypass surgery) can improve blood flow acutely, long-term patency remains a major limitation. Restenosis of vessels due to neointimal hyperplasia (NIH) causes failure rates to approach 50% at two years [7].

NIH is exacerbated by persistent vascular inflammation following acute injury and device implantation [8]. Resolution is an active physiologic process that terminates the acute phase of inflammation and promotes the clearance of proinflammatory factors and cellular debris to facilitate tissue repair. This process is orchestrated by signaling molecules including specialized pro-resolving lipid mediators (SPM) [9]. SPM are endogenously derived from polyunsaturated fatty acids (PUFA) and act on G-protein coupled receptors present on circulating cells, tissue macrophages and vascular cells [10, 11]. These interactions promote macrophage M2 polarization, limit the expression of adhesion molecules that promote leukocyte accumulation, and enhance the clearance of apoptotic cells and debris, ultimately leading to healing rather than chronic inflammation [12–14]. SPMs are derived from the PUFA precursors n-6 arachidonic acid (AA), n-3 docosapentaenoic acid (DPA), docosahexaenoic acid (DHA) and eicosapentaenoic acid (EPA). From these, the SPM derivative families include D- and E- series resolvins, protectins, maresins, and lipoxins, many of which have been produced via total organic synthesis [9, 15].

7S,8R,17S-trihydroxy-4Z,9E,11E,13Z,15E,19Z-docosahexaenoic acid (17S-RvD1 or RvD1) is an endogenous SPM derived from DHA. In prior work, we demonstrated pro-resolving effects of RvD1 on vascular cells *in vitro* and *in vivo* models of vascular injury. RvD1 attenuated human vascular smooth muscle cell (VSMC) proliferation and migration, attenuated endothelial cell (EC) p65 nuclear translocation, and reduced proliferation and migration in rat aortic VSMC [16, 17]. We previously demonstrated that perivascular administration of RvD1 safely facilitated resolution and attenuated neointimal hyperplasia in both a rat model of carotid angioplasty and a rabbit model of carotid bypass [17, 18].

Bioactive lipids such as RvD1 have poor shelf stability and are metabolized quickly in vivo; for example the half-life of RvD1 is in the order of hours due to the ubiquitous presence of dehydrogenases and reductases [19, 20]. Establishing adequate local bioavailability, and therefore clinical application to humans, can be challenging for lipid-based therapeutics. Our group has investigated formulations and devices to enhance local delivery of SPM to injured vessels

to promote repair and reduce NIH [21, 22]. 17R-RvD1 is a naturally occurring aspirin-triggered epimer of 17S-RvD1 which is more resistant to degradation by eicosanoid oxidoreductase [20]. 17R-RvD1 attenuated VSMC migration through cAMP and PKA mediated pathways, and oral administration of 17S-RvD1 reduced arterial wall inflammation following carotid angioplasty, but neither 17S-RvD1 nor 17R-RvD1 via oral route significantly attenuated intimal hyperplasia [23, 24].

Recently, several synthetic SPM analogues have been created that have simpler total organic synthesis and improved chemical stability. These include versions of lipoxin A4 (LXA4) analogues as well as an analogue of RvE1 (RX-10045) [25, 26]. Orr et al. recently described the synthesis of benzo-ring analogs of RvD1, including 17R/S-benzo-RvD1 (benzo-RvD1), that resist degradation by eicosanoid oxidoreductase and exerted a similar profile of pro-resolving biologic effects as RvD1 in mice and human phagocytes [19]. Thus, we sought to investigate the potential therapeutic effects of benzo-RvD1 on human vascular cells and in a rat model of carotid artery injury.

## Methods

### Reagents, cells and treatment protocol

Human greater saphenous veins discarded at the time of coronary or peripheral bypass grafting operations at The University of California- San Francisco (approved by the Institutional Review Board; UCSF Committee on Human Research- Number: 10–03395; the committee waived the need for informed consent) were used to prepare primary cell cultures of endothelial cells (EC) and vascular smooth muscle cells (VSMC), as described previously [27]. Human umbilical veins discarded at the time of delivery were utilized to harvest human umbilical vein endothelial cells (HUVECs), as previously described [28]. RAW 264.7 macrophages were purchased and utilized per vendor recommendations (ATCC).

The RvD1 analog, 17R/S-benzo-RvD1 (benzo-RvD1, $C_{22}H_{30}O_5$), was prepared via custom synthesis by Cayman Chemical (Ann Arbor, MI; Fig 1). Benzo-RvD1 physical properties were determined by UV spectrometry, LC-MS-MS, and gas chromatography/mass spectrometry (GC-MS). 17R-RvD1 was purchased from Cayman Chemical (Ann Arbor, MI).

### Cell migration assay

VSMC were grown to confluence on 24-well plates. After an overnight period of serum starvation, a standardized mechanical "scratch" wound was made across each well with a sterile 200-mL pipette tip. Detached cells were washed away with PBS, and remaining cells were exposed to fresh 0.5% DMEM with no agonist (negative control) or PDGF-BB (10 ng/mL; Sigma-Aldrich). 17R-RvD1 (0.1 nM– 100 nM, benzo-RvD1 (0.1 nM– 100 nM) or vehicle was added to appropriate wells just before treatment with agonist. Wounds were photographed with 10x power at baseline and at 16 hours, and wound closure was quantified by ImageJ. Each treatment was repeated in four wells and two 10x power fields were photographed per well.

For ECs, the bottoms of 24-well plates were coated with 1% gelatin and allowed to dry prior to plating. ECs were then grown to confluence on 24-well plates prior to an overnight period of relative serum starvation with 0.5% FBS media. The mechanical "scratch" was made in the same manner as with VSMC. Following this step, the remaining cells were exposed to fresh media with 0.5% FBS as a negative control or 10% FBS. 17R-RvD1 (1 nM– 100 nM), benzo-RvD1 (1 nM– 100 nM), or vehicle was added to appropriate wells with the addition of fresh media. Wounds were observed at baseline and at 16 hours, and wound closure was quantified by ImageJ.

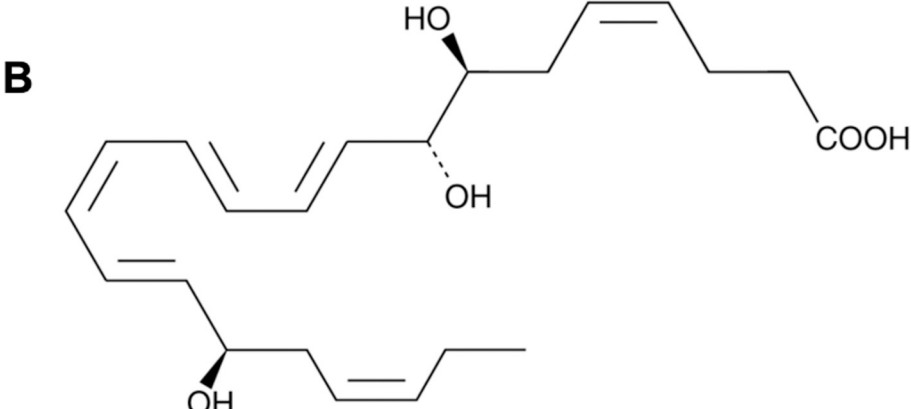

**Fig 1. Chemical structures of the RvD1 analogues.** A. Chemical structure of 17R/S-benzo-RvD1. B. Chemical structure of 17R-RvD1.

## Cell shape measurement

VSMCs were seeded onto 8-well chamber slides in 10% FBS DMEM for 24 hours, then serum-deprived overnight with DMEM containing 0.1% FBS. Cells were then pre-treated with either 17R-RvD1 or Benzo-RvD1 at 10nM and 100nM concentrations for 1 hour, followed by addition of PDGF-BB (10ng/ml) for 1 hour. Wells without SPM pretreatment and wells without PDGF-BB induction were included as positive and negative controls respectively. Cells were then fixed in 4% paraformaldehyde, permeabilized in 0.1% Triton X-100, stained with Alexa Fluor 568 phalloidin (Invitrogen) and mounted with DAPI-containing Fluoromount-G (SouthernBiotech). One wash with PBS was performed between steps. Each condition was tested in 4 separate wells. Images (20x) from 4 areas per well, were taken using fluorescence microscope and length to width ratios were measured for each well (20 cells per well) using Image J software.

## Cytotoxicity

Human VSMC and EC were grown to confluence on 24-well plates in 10% media and exposed to various concentrations of benzo-RvD1 (0.1–1000 nM). Cytotoxicity was quantified at 24 hours using a standard MTT assay (Sigma-Aldrich).

## Cell proliferation

Human VSMC and EC were seeded at a density of 800–1000 cells/200μL per well of a 96 well plate with full 10% serum media. After 24 hours of incubation, media was replaced with that

containing 10% serum and 10% alamarBlue and then incubated for 4 hours, after which base-line values were established using a fluorescence plate reader with excitation 560nm and emission 585nm. Cultures were then supplemented with 17R-RvD1 (0.1 nM– 100 nM), benzo-RvD1 (0.1 nM– 100 nM) or ethanol (vehicle) with 8 replicate wells for each condition. At days 4, 6, 8, 10, 12, 14 and 16, alamarBlue was added to wells and fluorescent readings were taken in the spectrophotometer. Treatments were then readministered in fresh media until the next time point. Wells without cells containing vitamin C (150 mg/mL) were used as positive control (rapid full reduction of alamarBlue) while media only was used as a negative control to subtract background fluorescence.

## Nuclear factor kB (NF-kB) activation

Human umbilical vein endothelial cells (HUVECs) were seeded onto 12 mm glass coverslips in 24 well plates at 75,000 cells/well. After adhering, cells underwent pretreatment with 17R-RvD1 (10 nM, 100 nM) or benzo-RvD1 (10 nM, 100 nM) or vehicle for 30 minutes, followed by addition of TNFα (1 ng/mL, Sigma-Aldrich) for 2 hours. Coverslips were then washed in ice-cold phosphate buffered saline (PBS), then fixed in 4% formaldehyde and treated with 0.5% Triton X-100 (Sigma-Alrich). Blocking with 2% FBS in 0.3% Triton X-100 was performed before overnight incubation at 4˚C with anti-p65 antibody (1:50; Santa Cruz Biotechnology, Dallas, TX). An Alexa Fluor 488 tagged secondary antibody (1:200, Life Technologies, Carlsbad, CA) was then used, followed by DAPI nuclear counterstaining. Images were taken with a fluorescence microscope and analyzed with ImageJ. NF-κB activation was quantified by the ratio of fluorescent intensity in the nuclear vs. cytoplasm.

## Macrophage phagocytosis

Stimulation of phagocytosis activity is a characteristic pro-resolving effect of SPM. Murine Raw 264.7 cells were plated on 96-well plates at a density of $5x10^5$ cells/ml. Cells were pretreated with either no treatment, LPS (100ug/ml), 17R-RvD1 (100-1000nM) and benzo-RvD1 (100-1000nM) for 1 hour, then either S. aureus (50:1 particle to macrophage ratio) or Zymosan (10:1 particle to macrophage ratio) bioparticles labeled with pHrodo (Invitrogen) were added for phagocytosis. Plates were analyzed using a microplate reader (SpectraMax M2; Molecular Devices) to detect the intensity of bioparticle fluorescence.

## Rat carotid artery angioplasty

Sprague- Dawley rats (350–500 g; N = 47) were used in compliance with an Institutional Animal Care and Use Committee-approved protocol (University of California, San Francisco # AN171612). Rats were anesthetized with buprenorphine, isoflurane, and lidocaine. Arterial injury was produced by balloon angioplasty of the left common carotid artery, as previously described [17, 29, 30]. Specifically, we cannulated the common carotid artery with a 2F balloon (Edwards Lifesciences, Irvine, Calif) inserted through the external carotid artery and inflated the balloon to 5 atm for 1 minute using a calibrated balloon inflation device (Boston Scientific, Marlborough, Mass). After angioplasty, the external carotid artery was ligated, and 25% pluronic gel loaded with either benzo-RvD1, 17R-RvD1 or ethanol (vehicle) was applied around the common carotid at the site of angioplasty. Rats were euthanized with isoflurane, carbon dioxide and bilateral thoracotomies, and arteries were harvested at POD3 for frozen section and immunostaining to analyze cellular proliferation and leukocytosis in angioplasty only (n = 3), vehicle (n = 4), and benzo-RvD1 (n = 4) treated arteries. For morphometric analysis, arteries were perfusion fixed and embedded in paraffin at POD14 in angioplasty only (n = 9), vehicle (n = 15), 17R-RvD1 (n = 6), and benzo-RvD1 (n = 6) treated animals.

## Pluronic gel preparations

A total of 200 ng of benzo-RvD1 or 17R-RvD1 was dissolved into 100 μL of 25% Pluronic F127 gel for delivery to arteries after angioplasty, allowing complete circumferential coverage of the injured artery. Vehicle gels were loaded with ethanol. Gels were kept on ice until perivascular application after angioplasty and subsequently allowed to solidify at body temperature, as previously described [27].

## Immunostaining

6-μm frozen sections of the harvested arteries were taken throughout the zone of injury. Frozen sections were fixed in acetone or 4% paraformaldehyde for 10 minutes before staining. Sections were then permeabilized with 0.2% Triton-X (Sigma-Aldrich). The sections were blocked in 10% goat serum. Blocking of endogenous avidin/ biotin was performed with a commercial kit (Vector, Burlingame, Calif) before incubation with one of the following primary antibodies: Ki67 (1:500; Abcam AB16667) and CD45 (1:150; Abcam AV10558). This was followed by incubation with a biotinylated goat-anti-rabbit secondary antibody (1:200; BioLegends, San Diego, CA). Conjugated streptavidin was applied to fluorescently label the secondary antibody (1:200; Dako). Fluorescently labeled sections were then mounted with DAPI mounting solution (SouthernBiotech).

Images of six sections per artery were obtained at 20x with a Nikon Widefield Epifluorescence microscope. Images were analyzed via ImageJ software. The intima/media was selected freehand, and the nuclei and antibody channels were separated. Areas of overlap between nuclei and antibody staining were designated as positive cells. The number of positive cells was normalized to total intima/media area.

## Morphometric analysis

After fixation in 4% formaldehyde and processing in 70% ethanol, specimens were paraffin embedded and 6-μm sections were taken throughout the zone of injury. Sections were stained with an elastin stain kit (Thermo Fisher Scientific, Waltham, Mass), and analysis was performed using ImageJ. Three sections within the zone of injury were analyzed for each specimen. Standard morphometric measurements were recorded, including luminal area, neointimal area (area inside the internal elastic lamina minus luminal area), and medial area (area inside the external elastic lamina minus area inside the internal elastic lamina). Vessels were harvested within the surgical plane, and periadventitial area was calculated as all tissue outside of the external elastic lamina.

## Statistical analysis

One-way ANOVA was utilized for initial multiple group analyses, followed by Tukey post-hoc tests for individual group comparisons. The threshold for statistical significance was set at $p < 0.05$.

# Results

## 17R-RvD1 and benzo-RvD1 reduce VSMC migration

A scratch assay was used to evaluate VSMC migration with PDGF-BB as a classic mitogen (n = 4). Pretreatment with benzo-RvD1 led to significant reduction in VSMC migration at concentrations as low as 0.1 nM (53%; $p < 0.01$) and did not demonstrate significant dose response across the tested range of 0.1–100 nM (Fig 2A). Pretreatment with 17R-RvD1 led to

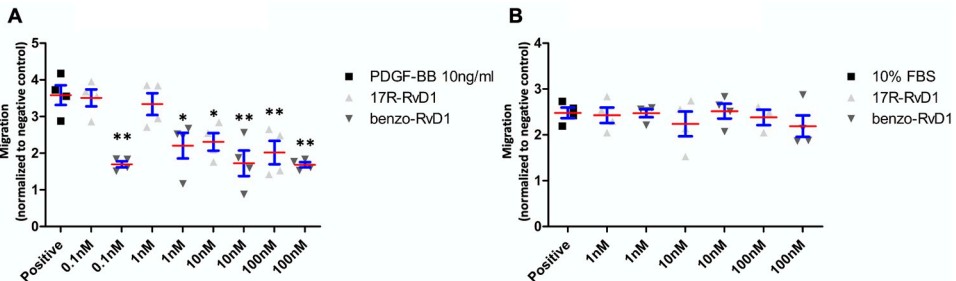

**Fig 2. 17R-RvD1 and benzo-RvD1 reduce VSMC migration.** A. Supplementation with Benzo-RvD1 or 17R-RvD1 led to significant reduction in VSMC migration 16 hours following PDGF-BB supplementation. ((*p<0.05, **p<0.01, n = 4). Bars represent SEM. B. Supplementation with Benzo-RvD1 or 17R-RvD1 had no effect on EC migration 16 hours following supplementation with 10% FBS. Red lines represent means and blue bars represent SEM.

similar inhibition of VSMC migration at 10 nM (36%; p<0.05) and 100 nM (44%; p<0.01) (Fig 2A).

In contrast to the observed effects of the compounds on VSMC migration, neither benzo-RvD1 nor 17R-RvD1 significantly impacted EC migration in the scratch assay (Fig 2B, n = 4).

## 17R-RvD1 and Benzo-RvD1 attenuate PDGF-BB induced VSMC cytoskeletal changes

PDGF exposure rapidly causes cytoskeletal rearrangement in VSMC, manifest as increased length to width ratio through stress fiber formation. Both 17R-RvD1 and Benzo-RvD1 (10nM, 100nM, n = 4 for all groups) significantly blunted this prototypic response, reducing the length to width ratio change by 55–58% compared to PDGF–BB only (p<0.01) (Fig 3A).

## 17R-RvD1 and benzo-RvD1 modestly inhibit VSMC and EC proliferation

Using a standard MTT assay, there was no evidence of cytotoxicity in either VSMC or EC across a range of Benzo-RvD1 exposures (0.1 nM– 1000 nM) (S1 Fig).

Benzo-RvD1 (0.1-100nM) and 17R-RvD1 (1-100nM) each modestly inhibited VSMC and EC proliferation without apparent dose effect (S2 Fig). For VSMC, this effect was statistically significant beginning at 0.1nM and above, ranging from 13–21% reduction in cell number at 13 days. For ECs, this effect was statistically significant at 1nM and above, ranging from 24–40% reduction in cell number at 8 days.

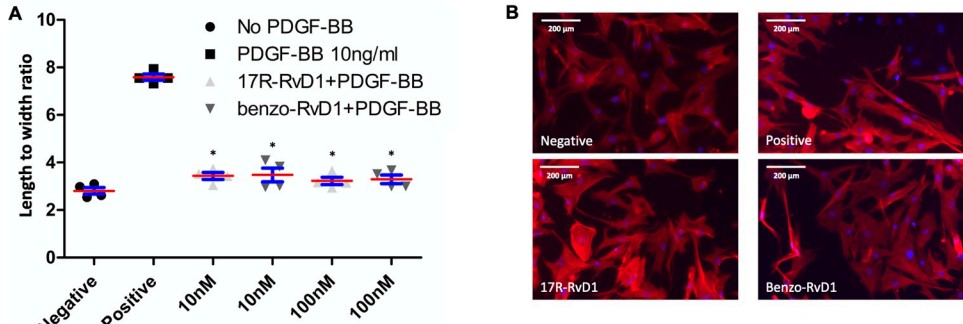

**Fig 3. 17R-RvD1 and benzo-RvD1 attenuate PDGF-BB induced cytoskeletal changes.** A. Pre-treatment with both 17R-RvD1 and Benzo-RvD1 attenuated PDGF-BB induced lengthening of VSMCs to nearly baseline levels. *p<0.01 compared to PDGF-BB only. Bars represent SEM. B. Representative images of VSMCs following staining with AlexaFlour 568 Phalloidin. Red lines represent means and blue bars represent SEM.

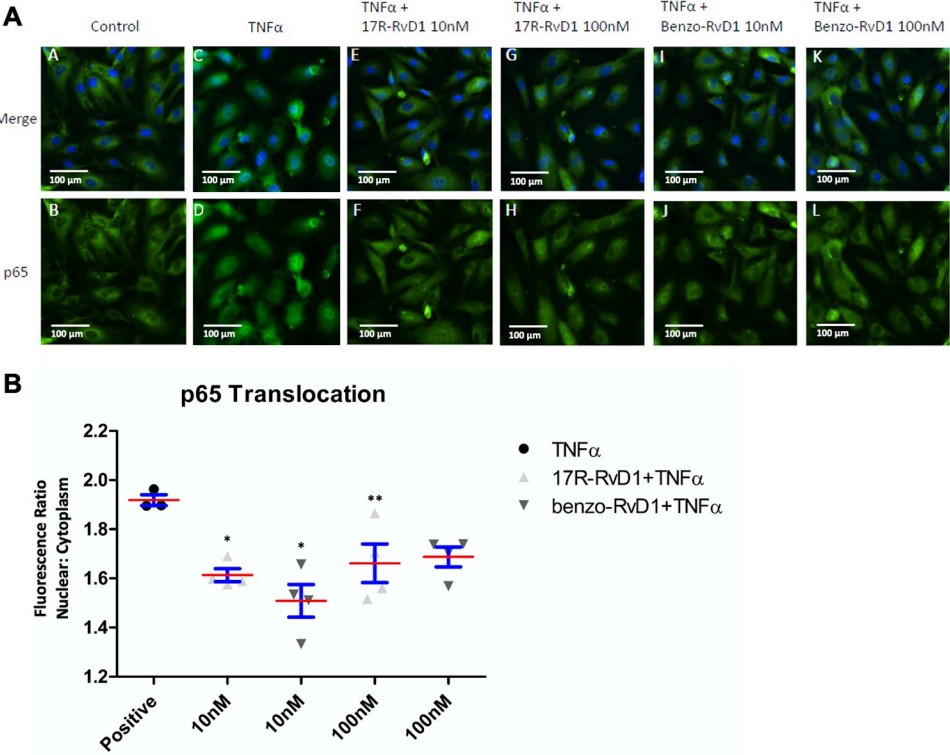

**Fig 4. 17R-RvD1 and benzo-RvD1 attenuate HUVEC p65 nuclear translocation.** The p65 nuclear translocation of endothelial cells is blunted with SPM supplementation. A. 17R-RvD1 and Benzo-RvD1 attenuated TNFa-stimulated p65 nuclear translocation (nuclear factor κB [NF-κB] activation) in human umbilical vein endothelialcells (HUVECs). Representative images are shown (A-L). B. Quantification of the nuclear:cytoplasmic p65 fluorescence ratio normalized to negative control. (*$P<0.05$, **$P<0.01$ compared to positive control, n = 4). Red lines represent means and blue bars represent SEM.

## 17R-RvD1 and benzo-RvD1 attenuate p65 translocation in EC

NF-κB activation is fundamental to the inflammatory response in many cell types, particularly vascular endothelium. Both 17R-RvD1 and benzo-RvD1 attenuated TNF-a induced p65 nuclear translocation in HUVECs. Pretreatment with 10 nM 17R-RvD1 inhibited TNF-induced p65 translocation in HUVECs by 16% ($p<0.05$). Similarly, benzo-RvD1 (10 nM) attenuated p65 translocation in HUVECs by 21% ($p<0.05$) (Fig 4A and 4B, n = 3–4).

## 17R-RvD1 and benzo-RvD1 enhance RAW 264.7 phagocytic activity

A fundamental property of SPM is stimulation of phagocytotic activity in neutrophils and macrophages. As expected, 17R-RvD1 and benzo-RvD1 significantly increased phagocytic activity of RAW 264.7 cells to S. aureus (bacterial) bioparticles compared to untreated cells. Pre-treatment with 17R-RvD1 and benzo-RvD1 resulted in 2x and 2.5x increased phagocytic activity compared to no treatment, respectively (Fig 5A; n = 3). Results were similar with Zymosan (sterile) bioparticles with 2.3x and 3.2x increased phagocytic activity (Fig 5B).

## 17R-RvD1 and benzo-RvD1 treatment did not impact early proliferation or leukocyte recruitment in injured rat carotid artery

Arteries harvested three days after balloon angioplasty were examined for cell proliferation (Ki67) and leukocyte recruitment (CD45). There was no visible staining for either antigen in

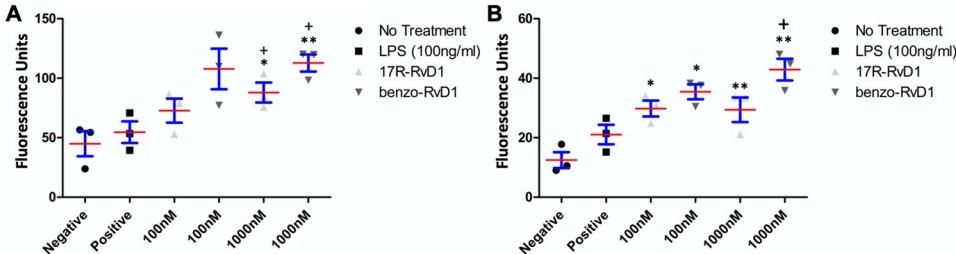

**Fig 5. 17R-RvD1 and benzo-RvD1 enhance phagocytic activity of Raw 264.7 cells.** 17R-RvD1 and benzo-RvD1 enhanced Raw 264.7 cell phagocytic activity compared to negative control of both S. aureus (A) and Zymosan bioparticles (B). (* $p<0.05$ compared to negative control. **$p<0.01$ compared to negative control. + $p<0.05$ compared to LPS. n = 3) Red lines represent means and blue bars represent SEM.

uninjured arteries. There were no significant differences noted between vehicle (n = 4) and benzo-RvD1 (n = 4) treated arteries with respect to Ki67 (15±7 vs 25±9% positive nuclei, p = 0.2) or CD45 (4±2 vs 7±3% positive nuclei, p = 0.3) indices.

## Perivascular application of 17R-RvD1 and benzo-RvD1 reduce NIH in injured rat carotid artery at 14 days

Local delivery of both benzo-RvD1 and 17R-RvD1 attenuated NIH at 14 days in the rat angioplasty model. Perivascular application of benzo-RvD1 gel was associated with reduced neointimal formation compared with the angioplasty-only group (0.08 mm$^2$ v. 0.18 mm$^2$; p<0.05) (Fig 5A). Benzo-RvD1 treatment decreased media formation compared with the angioplasty-only group (0.10 mm$^2$ v. 0.15 mm$^2$; p<0.01) and 17R-RvD1 reduced the neointima/media ratio compared to vehicle gel (0.63 v. 1.33; p<0.05), and these findings were similar to those observed for benzo-RvD1, but this was not statistically significant (0.81 vs 1.33; p = 0.58) (Fig 6B–6E). The predominant cell type in the developing neointima was alpha-SM actin (+) VSMC (Fig 6F).

## Discussion

Neointimal hyperplasia results from the acute inflammatory response following vascular intervention. When excessive, this process leads to clinical restenosis and treatment failure. Although progress has been made in drug elution technology utilizing agents such as paclitaxel and sirolimus, outcomes of vascular interventioins are still frequently hindered by restenosis. Moreover, paclitaxel in particular has cytotoxic properties that tend to retard, rather than promote, tissue healing [31]. Thus, there remains significant enthusiasm for the discovery of alternate approaches to reduce vascular injury/inflammation and enhance functional vascular repair following therapeutic interventions. SPM are novel bioactive lipid mediators that promote resolution, attenuate inflammation, and enhance tissue repair. These homeostatic properties are inherently attractive for vascular applications. However, significant translational hurdles remain including the development of optimal formulations and delivery methods. Our results demonstrate that the synthetic SPM analogue 17R/S- benzo-RvD1 attenuates VSMC proliferation and migration, inhibits TNF-induced NF-kB activation in EC, and enhances macrophage phagocytic activity. Perivascular treatment with benzo-RvD1, similar to the naturally occurring epimer 17R-RvD1, reduced NIH in the rat carotid angioplasty model. These data indicate that further investigation of novel SPM analogues to reduce vascular inflammation and downstream NIH should be pursued.

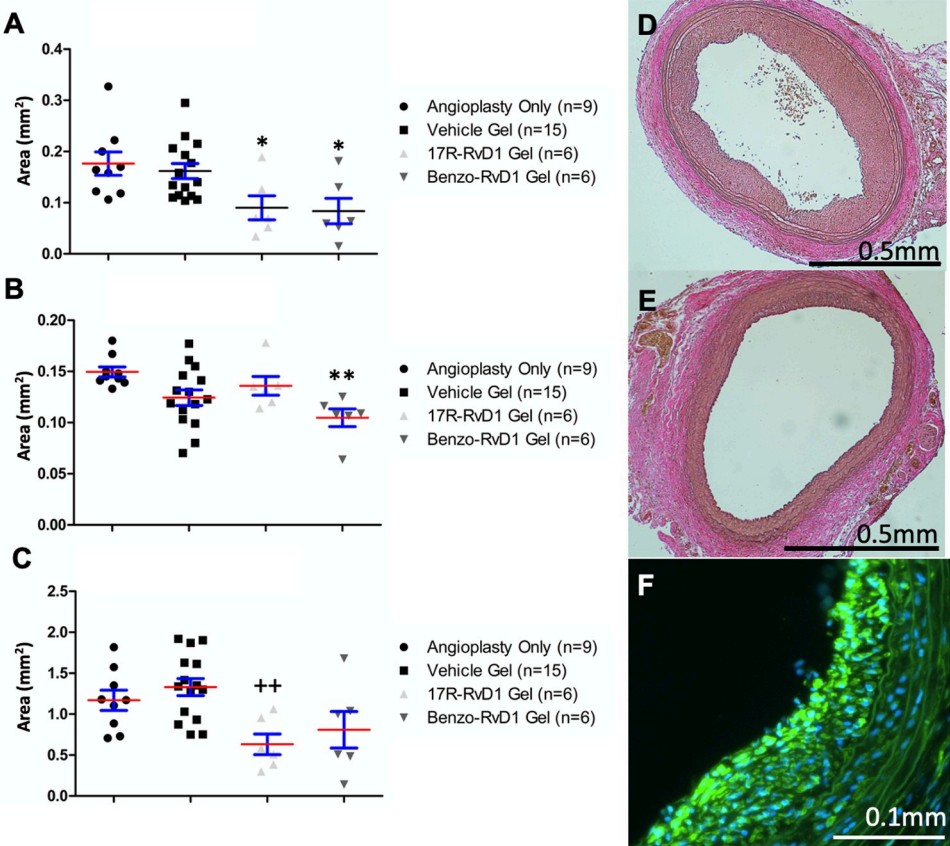

**Fig 6. 17R-RvD1 and benzo-RvD1 attenuate intimal hyperplasia in a rat model of carotid injury.** A. 17R-RvD1 and Benzo-RvD1 reduced neointimal area compared to angioplasty only. B. Benzo-RvD1 reduced medial area compared to angioplasty only. C. 17R-RvD1 reduced neointima:media area ratio compared to vehicle control group. ($^*$p<0.05 compared to angioplasty only, $^{**}$p<0.01 compared to angioplasty only, ++p<0.01 compared to vehicle, angioplasty only n = 9, vehicle n = 15, 17R-RvD1 n = 6, Benzo-RvD1 n = 6). Red lines represent means and blue bars represent SEM. D. Representative section of vehicle treated artery. E. Benzo-RvD1 treated artery. F. α-SM actin staining demonstrating neointimal VSMC in vehicle treated artery.

While we have previously demonstrated the beneficial vascular effects of the endogenous SPM 17S-RvD1 [17, 18], there are potential advantages offered by other epimers and novel, synthetic analogues of bioactive lipid mediators. Benzo-RvD1, a synthetic analogue, offers the potential for improved shelf life and prolonged bioactivity due to increased stability which resists degradation by dehydrogenases and oxidoreductases. Our first objective was to demonstrate that RvD1 analogues have similar physiologic effects on vascular cells and macrophages fundamental to the repair and remodeling process.

VSMC proliferation and migration are fundamental to the neointimal hyperplasia response that follows vascular injury and inflammation [32]. At the cellular level, both 17R-RvD1 and benzo-RvD1 affected a modest reduction in VSMC proliferation and notable inhibition of PDGF-stimulated VSMC migration. In contrast these compounds demonstrated no significant effect on EC migration. As previously shown for other SPM [24, 33], both 17R-RvD1 and benzo-RvD1 attenuated PDGF-induced VSMC cytoskeletal rearrangement, a critical initial step in the motogenic response. We suspect that inhibition of VSMC migration is a primary mechanism underlying the vasculoprotective properties of several SPM including RvD1, RvD2, RvE1, MaR1 and LxA4 [16, 17, 33–35].

Macrophage M2 polarization is an important component of resolution following acute inflammation, as it enhances phagocytic activity and encourages the clearance of apoptotic debris and proinflammatory cytokines. Prior studies have demonstrated M2 polarization of macrophages following exposure to a variety of SPMs [12, 13, 36, 37]. As an indirect measure, we demonstrate that benzo-RvD1 and 17R-RvD1 markedly enhanced phagocytic activity of RAW 264.7 cells to both S. aureus and zymosan particles; S. aureus represent bacterial or infectious particles while zymosan represents sterile debris. This confirms the pro-resolving activity of these RvD1 analogues.

At the level of intracellular cell-signaling, NF-kB is a major transcription factor that upregulates genes associated with inflammation. p65 is one of the five components that comprise NF-kB, and it is typically involved in the inflammatory response. Importantly, it is also implicated in VSMC proliferation, migration, and oxidative signaling pathways [33, 38]. In our study, both SPMs modestly inhibited TNF-stimulated p65 nuclear translocation in HUVECs.

Similar to our prior reports using perivascular administration of 17(S)-RvD1, we observed significant attenuation of NIH by benzo-RvD1 and 17R-RvD1 in the rat carotid balloon angioplasty model. Neointimal area was reduced by approximately 50% compared to untreated and vehicle-treated controls. These data support further potential development of these SPM analogues as candidate anti-restenosis therapeutics.

## Limitations

Our study was designed as proof-of-concept for evaluation of these analogues and has several limitations. Although we demonstrated reduced NIH, we are unable to fully explain the cellular mechanisms for the observed *in vivo* effects of benzo-RvD1. In prior studies we found that local delivery of 17S-RvD1 using a biodegradable thin film reduced early proliferative index, oxidative stress, and NF-kB activation in injured rat carotid arteries [17]. We published similar findings in a rabbit carotid bypass model using 17S-RvD1. Differences in local pharmacokinetics between gel and film approaches, variability across the animal models, as well as compound-specific differences, could be in play. We suspect that the consistently observed in-vitro effects of these SPM on VSMC migration are of central importance to the attenuation of NIH, but cell migration cannot be readily quantified in live animals or tissue samples.

The rat model of carotid artery angioplasty was chosen for this study due to its common use and relative cost-effectiveness for proof-of-concept. However, this is a non-atherosclerotic/atherogenic model, and the results may not be applicable to other circumstances of restenosis such as venous or synthetic grafts used for vascular bypass. The tissue concentrations of SPM were not measured, and pharmacokinetic requirements for these compounds in this application are unknown. Therefore, although a relatively large dose of SPM was loaded onto the gels for treatment, the efficiency and duration of SPM delivery to the arterial wall is unclear. Further studies exploring dosing range, longer time points, and higher-level animal models are needed to better characterize the effects of local application of these analogues in acute vascular injury.

## Conclusion

Perivascular delivery of 17R-RvD1 and the synthetic analogue 17-R/S-benzo-RvD1 significantly attenuated NIH in a rat model of carotid angioplasty, with no signs of local toxicity. Benzo-RvD1 is a novel synthetic SPM with anti-inflammatory and pro-resolving properties that appear relevant for a candidate anti-restenosis compound. Further studies are indicated to better define the mechanism of SPM effects in vascular injury, and the potential preclinical development of SPM analogues for anti-restenosis strategies.

## Supporting information

**S1 Fig. Benzo-RvD1 is not toxic to VSMC and EC.** Benzo-RvD1 (0.1-1000nM) did not demonstrate significant cytotoxicity in endothelial cells (EC) and vascular smooth muscle cells (SMC) on a standard MTT assay.
(TIF)

**S2 Fig. 17R-RvD1 and benzo-RvD1 modestly inhibit VSMC and EC proliferation.** Treatment of VSMCs with Benzo-RvD1 and 17R-RvD1 leads to a significant, modest reduction in cellular proliferation across a wide range of concentrations and time points ($^* <0.05$; $^{**} <0.001$, n = 4 all timepoints). Treatment of ECs with Benzo-RvD1 and 17R-RvD1 leads to a significant, modest reduction in cellular proliferation only at later time points. ($^* <0.05$; $^{**} <0.001$, n = 4 all timepoints). Bars represent SEM.
(TIF)

**S1 Data.**
(XLSX)

## Author Contributions

**Conceptualization:** Evan C. Werlin, Michael S. Conte.

**Data curation:** Alexander S. Kim, Evan C. Werlin, Hideo Kagaya, Giorgio Mottola, Masood Jan.

**Formal analysis:** Alexander S. Kim, Evan C. Werlin, Hideo Kagaya, Giorgio Mottola.

**Funding acquisition:** Evan C. Werlin, Michael S. Conte.

**Investigation:** Alexander S. Kim, Evan C. Werlin, Bian Wu, Giorgio Mottola.

**Methodology:** Alexander S. Kim, Evan C. Werlin, Hideo Kagaya, Mian Chen, Bian Wu, Giorgio Mottola, Michael S. Conte.

**Project administration:** Michael S. Conte.

**Writing – original draft:** Alexander S. Kim, Evan C. Werlin.

**Writing – review & editing:** Alexander S. Kim, Hideo Kagaya, Mian Chen, Bian Wu, Giorgio Mottola, Masood Jan, Michael S. Conte.

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
