## [Decision Letter · Decision Letter 0]

19 Oct 2021

PONE-D-21-2424517R/S-Benzo-RvD1, a synthetic resolvin D1 analogue, attenuates neointimal hyperplasia in a rat model of acute vascular injuryPLOS ONE

Dear Dr. Conte,

Thank you for submitting your manuscript to PLOS ONE. After careful consideration, we feel that it has merit but does not fully meet PLOS ONE’s publication criteria as it currently stands. Therefore, we invite you to submit a revised version of the manuscript that addresses the points raised during the review process.

We look forward to receiving your revised manuscript.

Kind regards,

Michael Bader

Academic Editor

PLOS ONE

2. To comply with PLOS ONE submissions requirements, in your Methods section, please provide additional information on the animal research and ensure you have included details on (1) methods of sacrifice, (2) methods of anesthesia and/or analgesia.

“I have read the journal's policy and the authors of this manuscript have the following competing interests:

MSC - co-Founder of VasaRx and co-inventor of IP related to this work with the Regents of University of California and Brigham and Women's Hospital”

Additional Editor Comments (if provided):

Reviewers' comments:

Reviewer's Responses to Questions

**Comments to the Author**

1. Is the manuscript technically sound, and do the data support the conclusions?

Reviewer #1: Yes

2. Has the statistical analysis been performed appropriately and rigorously? 

Reviewer #1: Yes

3. Have the authors made all data underlying the findings in their manuscript fully available?

Reviewer #1: Yes

4. Is the manuscript presented in an intelligible fashion and written in standard English?

Reviewer #1: Yes

5. Review Comments to the Author

Reviewer #1: The paper proposed by Kim AS et al is titled : « 17R/S-Benzo-RvD1, a synthetic resolvin D1 analog, attenuates neointimal hyperplasia in a rat model of acute vascular injury ».

The authors demonstrated that neointimal hyperplasia (NIH) is attenuated by RvD1 treatment following acute arterial injury in rats.

The authors adopted an interesting translational approach to describe their message.

They used VSMC, endothelial cells, HUVEC, and an in vivo model of induced intimal hyperplasia in rats.

Their multidisciplinary methods include migration assay, cell shape measurement, cytotoxicity, cell proliferation, immunostaining, in vivo analyses.

Although the paper is well designed and written, some minor concerns need to be addressed before acceptance.

1- Add a reference for the two first sentences of the introduction.

2- In the statistical analysis paragraph of the Material and methods section, add information about the minimum p from which data were considered significant. For example, p<0.05.

3. In multiple group analyses, one way ANOVA followed by Bonferroni or Tuckey post-hoc tests may better apply in this context. The authors can either repeat the appropriate statistics tests when applicable, or justify their choice in a limitation section.

4. As the `n` are low, for transparency, the authors are invited to present all bar graphs as scatter/dot plot graphs, showing mean’s and SEM’s bars. (except graphs shown in supplementary figures, as scatter/dot plot may be less readable than bar graphs).

5. Figure 3 and 4, add a scale for the pictures.

6. Add a limitation section after the discussion and before the conclusion to address the main challenges met by this study, the limitations, and possible ways to overcome them in future investigations. This can include justification of the small number of samples n = 4 or 3; feasibility of further experiments or analyses (qPCR, western blot, etc).

7. Add a Highlight section with up to 5 bullet points to describe the main findings of the paper or/and patient-oriented potential applications.

6. PLOS authors have the option to publish the peer review history of their article (what does this mean?). If published, this will include your full peer review and any attached files.

Reviewer #1: **Yes: **Roddy Hiram

---

## [Author Response · Author response to Decision Letter 0]

11 Dec 2021

Michael Bader, Academic Editor

PLOS ONE

Re: PONE-D-21-24245

17R/S-Benzo-RvD1, a synthetic resolvin D1 analogue, attenuates neointimal hyperplasia in a rat model of acute vascular injury

Dear PLOS-One Reviewers,

Thank you for reviewing our submitted work. Below are our responses to the comments and questions from the review:

Responses to general guidelines

1. Edited manuscript to meet PLOS ONE’s style requirements, including the file name

2. Added (1) methods of sacrifice, (2) methods of anesthesia and analgesia

3. Included statement “This does not alter our adhered to PLOS ONE policies on sharing data and materials” and included updated Competing Interests statement on cover letter

4. Included ORCID iD for Michael S. Conte

5. Included supplemental figure data pertinent to the section that previously stated “data not shown” (toxicity) proliferation figure is now supplemental figure 2. 

Responses to individual reviewer questions/comments

1. Add a reference for the two first sentences of the introduction

a. Added references for the two first sentences of the introduction

2. In the statistical analysis paragraph of the Material and methods section, add information about the minimum p from which data were considered significant. For example, p<0.05.

a. Added information regarding the minimum p from which data were considered significant, which was p<0.05

3. In multiple group analyses, one way ANOVA followed by Bonferroni or Tuckey post-hoc tests may better apply in this context. The authors can either repeat the appropriate statistics tests when applicable, or justify their choice in a limitation section.

a. We performed ANOVA with post-hoc Tukey for individual group comparisons and edited the manuscript text to reflect this.

4. As the `n` are low, for transparency, the authors are invited to present all bar graphs as scatter/dot plot graphs, showing mean’s and SEM’s bars. (except graphs shown in supplementary figures, as scatter/dot plot may be less readable than bar graphs).

a. All bar graphs (with exception of supplemental figures) were switched to scatter plots with means and SEM bars.

5. Figure 3 and 4, add a scale for the pictures.

a. Figure 3 and 4 – added scale bars

6. Add a limitation section after the discussion and before the conclusion to address the main challenges met by this study, the limitations, and possible ways to overcome them in future investigations. This can include justification of the small number of samples n = 4 or 3; feasibility of further experiments or analyses (qPCR, western blot, etc).

a. Added a limitation section following the discussion

7. Add a Highlight section with up to 5 bullet points to describe the main findings of the paper or/and patient-oriented potential applications.

a. Highlight section added with 3 bullet points to describe the main findings of the paper

Thank you for considering our revised manuscript for publication. We look forward to any additional comments or questions.

Sincerely,

Michael S. Conte MD

Alexander Kim MD

---

## [Editor Report · Decision Letter 1]

7 Feb 2022

17R/S-Benzo-RvD1, a synthetic resolvin D1 analogue, attenuates neointimal hyperplasia in a rat model of acute vascular injury

PONE-D-21-24245R1

Dear Dr. Conte,

We’re pleased to inform you that your manuscript has been judged scientifically suitable for publication and will be formally accepted for publication once it meets all outstanding technical requirements.

Kind regards,

Michael Bader

Academic Editor

PLOS ONE
---

## [Editor Report · Acceptance letter]

18 Feb 2022

PONE-D-21-24245R1 

17R/S-Benzo-RvD1, a synthetic resolvin D1 analogue, attenuates neointimal hyperplasia in a rat model of acute vascular injury 

Dear Dr. Conte:

I'm pleased to inform you that your manuscript has been deemed suitable for publication in PLOS ONE. Congratulations! Your manuscript is now with our production department. 

Kind regards, 

on behalf of

Prof. Michael Bader 

Academic Editor

PLOS ONE